# Understanding Compositionality in Data Embeddings

## Abstract

Embeddings are used in AI to represent symbolic structures such as knowledge graphs. However, the representations obtained cannot be directly interpreted by humans, and may further contain unintended information. We investigate how data embeddings might incorporate such information, despite that information not being used during the training process. We introduce two methods: (1) Correlation-based Compositionality Detection, which measures correlation between known attributes and embeddings, and (2) Additive Compositionality Detection, a process of decomposing embeddings into an additive composition of individual vectors representing attributes. We apply our methods across three domains: word embeddings using word2vec, which is based on a shallow, two-layer neural network model; sentence embeddings using SBERT, which uses a transformer architecture; and knowledge graph embeddings. We show that word embeddings can be interpreted as composed of semantic and morphological information, and that sentence embeddings can be interpreted as the sum of individual word embeddings. In the domain of knowledge graph embeddings, our methods show that attributes of graph nodes can be inferred, even when these attributes are not used in training the embeddings. Our methods are an improvement over previous approaches for decomposing embeddings in that our methods are 1) more general: they can be applied to multiple embedding types; 2) provide quantitative information about the decomposition; and 3) provide a statistically robust metric for determining the decomposition of an embedding.

## 1 Introduction

In AI research, embeddings are used to represent symbolic structures such as *knowledge graphs* as collections of vectors of fixed dimension. By converting to embeddings, standard algebraic techniques can be used to perform inferences on symbolic data. In other words, using embeddings allows for a convenient way to model and process data. This paper examines the extent to which vector embeddings can decomposed into different informative signals, and how those signals can be disentangled and interpreted.

Knowledge graphs are a way of encoding explicit declarative knowledge about a set of entities in a domain and the relations between those entities. They are a powerful tool to capture structured information about the world and model complex relationships between various entities. With the rise of massive knowledge bases and the need for efficient querying and inference, traditional symbolic reasoning on knowledge graphs can become computationally expensive.

To address these challenges, *graph embeddings* have been introduced as a method to convert the structured information of knowledge graphs into a continuous vector space. These embeddings aim to capture the topological relations and semantic meanings of entities and relationships in the graph. The conversion of symbolic constructs such as knowledge graphs into continuous embeddings enables efficient algebraic operations, similarity calculations, and other tasks. For instance, in bipartite graph representations, graph embeddings can reflect properties like a user liking a certain movie. The efficiency and expressiveness of these embeddings have proven useful across many applications, including link prediction (which we focus on here), node classification (Ji et al., 2021), and graph generation (Bo et al., 2021).

Many problems can naturally be cast in a knowledge graph setting, by defining the entities and the relation(s) between them. For example, the standard technique known as *word embedding* defines the entities as words,

and the relation between words as one of "co-occurrence", such that two words are related if they often occur in the vicinity of one another. In this and in many other cases, the strength of the relation is used too, and can be represented as a weight on the edge of the graph.

However, a challenge arises: these embeddings, drawn from real-world data to encode either graph topological or word context relations, may not always be transparent to human interpretation. Attempting to interpret embeddings in a compositional way implies that an embedding can be decomposed into distinct information components. However, this opacity makes potential unintended information hard to detect and assess, further complicating our understanding of how different components merge within the embedding space.

We are interested in the compositional properties of vector embeddings, namely: do they represent relations in such a way that vector manipulations such as addition can be interpreted as linguistic inflection, word composition or collections of attributes? A big open question in using embeddings to represent structured entities is whether the relations between them can be encoded as simple vector operations such as addition. In this paper, we evaluate whether vector representations of structured entities such as words, sentences, or users, can indeed be decomposed additively. In the case of sentences, we choose instances of the Transformer architecture as model, since this is an extremely successful and popular architecture, and showing that compositionality is additive in that model contributes to explainability. For word embeddings, we look at word2vec as a static model that is still widely used and different from the Transformer model. We will present two methods which can be applied to already trained embeddings and provide a quantitative description of the decomposition of embeddings.

**Our Methods** Our work is most aligned with that of Andreas (2019); Hewitt & Manning (2019); Bose & Hamilton (2019). We are interested in the extent to which embeddings can be additively decomposed into component parts. We examine three different kinds of data embedding: 1) word embeddings, 2) sentence embeddings, and 3) knowledge graph embeddings.

In the example of word embedding, we use pretrained Word2vec (Mikolov et al., 2013a) embeddings and investigate the extent to which these word embeddings can be analysed as a composition of their semantic meaning and their syntactic structure. In the example of sentence embeddings, we use sentence embeddings from BERT (Devlin et al., 2018), specifically, the CLS token embedding, and look at the extent to which simple sentences may be analysed as an additive compositionality of their constituent words. Finally, we look at knowledge graph embedding. In this problem, we train a set of embeddings over the MovieLens dataset (Harper & Konstan, 2015). This dataset contains entities for users and entities for movies, and relations on the knowledge graph consist of the users' ratings of the movies. We train our embeddings with the objective of performing *link prediction*, that is, the task of predicting whether a link holds between two entities. We describe this in more detail in section 2, however, the key point is that we learn the embeddings *without any reference to the demographic attributes of the users*, e.g. gender or age. We investigate the extent to which the user embeddings are in fact composed of an additive compositionality of demographic attributes, even though these are not used in training.

Throughout the three problems described above, we ask whether we can decompose an embedding into interpretable components. Specifically, we investigate additive compositionality, that is of the type $\phi(x) = \phi(x1) + \phi(x2)$.

**Approach** We introduce two distinct methods to analyse the extent to which embeddings can be interpreted as a composition of interpretable components.

1. **Correlation-based Compositionality Detection** We use Canonical Correlation Analysis (CCA) to provide a novel approach to measure the correlation between interpretable attributes and the data embedding itself. This method provides a quantitative measure of compositionality.

2. **Additive Compositionality Detection** We treat embeddings as additive compositionality of meaningful vector directions. We view an embedding $v$ as an aggregated sum $v = x_1 + x_2 + \ldots + x_k$, with each component $x_i$ a distinct meaningful direction within the vector space that represents an attribute (such as gender, age, etc.).

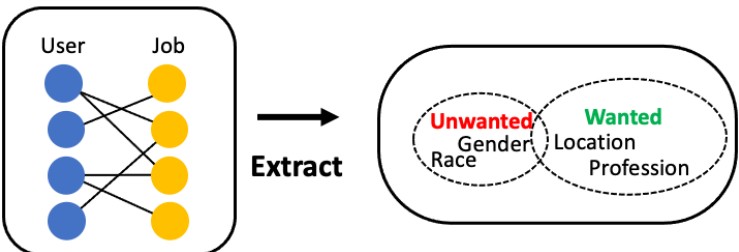

Figure 1: Embedding contains an information of both wanted and unwanted information

**Improvements Over Previous Approaches**   Unlike earlier models, our methods are versatile across different embedding types. We concentrate on additive compositionality and introduce two novel methods to systematically measure this additive compositionality. We also explore similar properties in knowledge graph embeddings. Unlike previous methods that utilised MLPs or feedforward neural networks to detect these attributes in knowledge graph embeddings, our study emphasizes that addition, rather than complex neural architectures, effectively captures the composition of meaning in different kinds of embeddings. Approaches such as Shwartz & Dagan (2019) Mikolov et al. (2013b) consider only how word embeddings should be decomposed. Similarly, Bose & Hamilton (2019) Fisher et al. (2020) consider only the interpretation of graph embeddings. Here, we show that the same methods can be used across different embedding types.

While Mikolov et al. (2013b); Bose & Hamilton (2019) show that embeddings can be decomposed into simple attributes, they only provide a qualitative decomposition, whereas we are able to provide a weighting that quantifies how much each component contributes to the overall compositionality of attributes by the correlation-based compositionality detection.

Furthermore, our Additive Compositionality Detection method provides a novel way to detect signal compositionality in embeddings. We consider an embedding $v$ as a cumulative sum given by $v = x_1 + x_2 + \cdots + x_k$, where each $x_i$ denotes a unique direction in the vector space corresponding to attributes. This was already done implicitly by Mikolov et al. (2013b), however, we provide a systematic method by which to isolate signals in the vector space and confirm the robustness of these signals via statistical testing.

**Findings**   We apply our methods to word embeddings, sentence embeddings, and graph embeddings. We find that word embeddings can be decomposed into semantic and morphological components. Similarly, for BERT sentence embeddings, we find that the sentence embeddings can be decomposed into a sum of individual word embeddings. Finally, we show that embeddings corresponding to users in a database of users and movie ratings can be decomposed into a sum of embeddings corresponding to demographic attributes such as gender, age, and so on, *even though these attributes are not used in the training of the embeddings.*

## 1.1   Related Work

There has been a wide range of research into how composition is encoded in embeddings produced by word, sentence, and graph embedding models. A classic illustration (Mikolov et al., 2013b) is the relationship between the embeddings of the words "King" and "Queen":

$$\mathbf{x}_{king} - \mathbf{x}_{man} + \mathbf{x}_{woman} \approx \mathbf{x}_{queen}$$

This provides the possibility to perform analogical inferences, where we can predict relationships (such as gender) between words.

**Compositionality in Word Embeddings**   There have been a number of approaches to understanding compositionality in word embeddings. Mikolov et al. (2013a) show that words can be decomposed semantically (as in the example above) and also morphologically - for example $\mathbf{x}_{quickly} - \mathbf{x}_{quick} + \mathbf{x}_{slow} \approx \mathbf{x}_{slowly}$. We extend this observation to multiple suffixes.

A further approach is Disentangled Representation Learning (DRL) (Bengio et al., 2013), which detects and separates attributes within data embeddings. Such disentangled representations, which can be deconstructed into components, enhance the explicability of the models trained. Each constituent in the latent space pertains to a discrete attribute or feature, thereby simplifying manipulation and control of data representations.

Shwartz & Dagan (2019) undertook an examination of word representation compositionality via six tasks, probing into the phenomena of semantic drift and implicit meaning. Andreas (2019) postulated a metric for compositionality based on the approximation fidelity of observed representations when assembled from inferred primitives.

Word embeddings have further been examined for presence of biases (Bolukbasi et al., 2016). Biases in data embeddings can inadvertently reflect societal norms and prejudices. For instance, associations in word embeddings often reveal embedded gender biases (Jonauskaite et al., 2021; Sutton et al., 2018; Caliskan et al., 2017). The methods we propose could, in future work, be used to identify and potentially remove bias.

**Compositionality in Sentence Embeddings**    There has been a wide range of research examining lexical composition in sentence embedding models. Much of this work is focussed on the presence or otherwise of tree-like structures representing syntax in models.

While BERT does not have explicit syntactic trees during training, the representations it learns capture significant syntactic information (Hewitt & Manning, 2019). Ettinger et al. (2016) developed a dataset to identify semantic roles in embeddings, such as whether "professor" is the agent of "recommend". They also looked at semantic scope by altering sentence meanings without much lexical change. Dasgupta et al. (2018) created a dataset examining word combinations in embeddings. They modified sentences to study natural language inference relations, involving changes like word order and addition of words like "more/less" or "not".

Adi et al. (2016) presented three evaluation techniques for sentence embeddings: measuring sentence length, identifying a word in a sentence, and determining word order. In tests, LSTM auto-encoders performed well in the latter two tasks.

Probing tasks and representational similarity analysis (RSA) have been used to investigate model capabilities in encoding linguistic features and compositional meaning. Studies such as Lepori & McCoy (2020) and Chrupała & Alishahi (2019) apply RSA to map neural activations to symbolic structures like syntax trees. Other works like Klafka & Ettinger (2020) and Ettinger et al. (2018) use probing tasks to examine model understanding of sentence composition and linguistic dependencies. Additionally, Tenney et al. (2019) introduces edge probing tasks to examine how models like ELMo and BERT encode sentence structure, emphasizing their capabilities in representing syntactic information while noting modest improvements in semantic tasks over non-contextual baselines.

RNNs and transformers implicitly encode complex symbolic structures. Soulos et al. (2020) illustrates how recurrent neural networks encode symbolic structures effectively, while Yu & Ettinger (2020) discusses the limitations and capabilities of transformers in handling nuanced compositionality.

In our work, we investigate a very straightforward form of compositionality—additive—finding that it is surprisingly effective at representing at least simple sentences. This finding contributes to explainability of how Transformer-based models represent linguistic composition.

**Compositionality in Graph Embeddings**    While the concept of compositionality has been deeply studied in fields like linguistics, there is less work in graph embeddings. Much of the work is focussed on the problem of debiasing embeddings (Bose & Hamilton, 2019; Chen et al., 2013; Zemel et al., 2013; Zhu et al., 2015; Wu et al., 2016; Fisher et al., 2020).

We can learn a word's semantic content from the distribution of word frequencies in its context. However, it has been observed that these distributions contain also information of different nature, including associations and biases that reflect customs and practices. For example it is known that the embeddings of colour names are not gender neutral, nor are those of job titles or academic disciplines. For example, engineering disciplines

and leadership jobs may tend to be represented in a "more male" way than artistic disciplines or service jobs (Jonauskaite et al., 2021; Sutton et al., 2018; Caliskan et al., 2017).

Similar biases have been shown to be present in Knowledge Graph embeddings. Recent work such as Fisher et al. (2020) Bose & Hamilton (2019) use adversarial loss to train the model neutral to sensitive attributes. Such a bias can also be observed in movie recommender systems whose embedding is simply trained from a set of movie ratings. This work discusses new ways to detect it.

**Compositionality in Deep Learning More Generally**  The ability for deep neural networks to reason compositionally is desirable and research has been carried out across a range of architectures. Kim & Linzen (2019) compares how Gated Recurrent Units (GRUs) and Simple Recurrent Networks (SRNs) manage compositional generalization, and Wu et al. (2020) analyzes the similarities and differences in contextual word representation models across distinct architectures.

Research is increasingly focused on assessing and improving how well neural models generalize to unseen compositions. For example, Lake & Baroni (2018) and Kim & Linzen (2020) examine the limitations of RNNs, LSTMs, and Transformers in handling novel, systematic tasks, highlighting the need for models that can better mimic human-like generalization. These findings underscore a significant gap in current models' abilities to extrapolate beyond trained examples. In response, Lake & Baroni (2023) develops a meta-learning approach to better equip neural networks with the capability for systematic generalization, aiming to emulate more human-like performance in compositional tasks. Hupkes et al. (2020) introduces a set of task-independent tests to evaluate the compositional generalization of neural models against linguistic and philosophical theories, applying these tests to sequence-to-sequence models on the PCFG SET dataset, revealing distinct strengths and weaknesses in recurrent, convolution-based, and transformer architectures. There are also some other studies investigate the structural and functional aspects of neural networks to uncover how different architectures contribute to or hinder compositional understanding (Mu & Andreas, 2020; Lepori et al., 2023).

## 1.2 Structure of Paper

Section 2 covers embedding, mapping elements to vector spaces, focusing on word, sentence, and graph embeddings. Section 3 introduces two methods: *Correlation-based Compositionality Detection* and *Additive Compositionality Detection* to detect the composition of signals in data embeddings. Section 4 presents experiments on three data embeddings, and Section 5 discusses results.

## 2 Mathematical Preliminaries

In machine learning, embedding is the process of mapping elements from a set, denoted as $I$, to points in a vector space. We write a set of coordinates $\mathbf{B}$ to represent the items of $I$ as follows:

$$\mathbf{B} = \Phi(I)$$

where $\Phi$ is the mapping function that maps the items (elements of the set) to their coordinates. This embedding function can be learned from a set of data containing those items: for words, this can be done by exploiting co-occurrence statistics between words; for elements of a graph, by exploiting the topology, i.e., the relations between different elements.

In the example of word embeddings and knowledge graph embedding we will make use of co-occurrence or relational information to create the embedding. In the example of sentence embedding we will make use of the CLS token from BERT. In both cases we will be interested how the embeddings of structured objects (e.g. sentences) can depend on the relations between those structures.

## 2.1 Word Embedding

**Word2Vec**  word2vec, as introduced by Mikolov et al. (2013a), is a method to embed words into vectors based on the distributional hypothesis: words in similar contexts have similar meanings. It consists of two

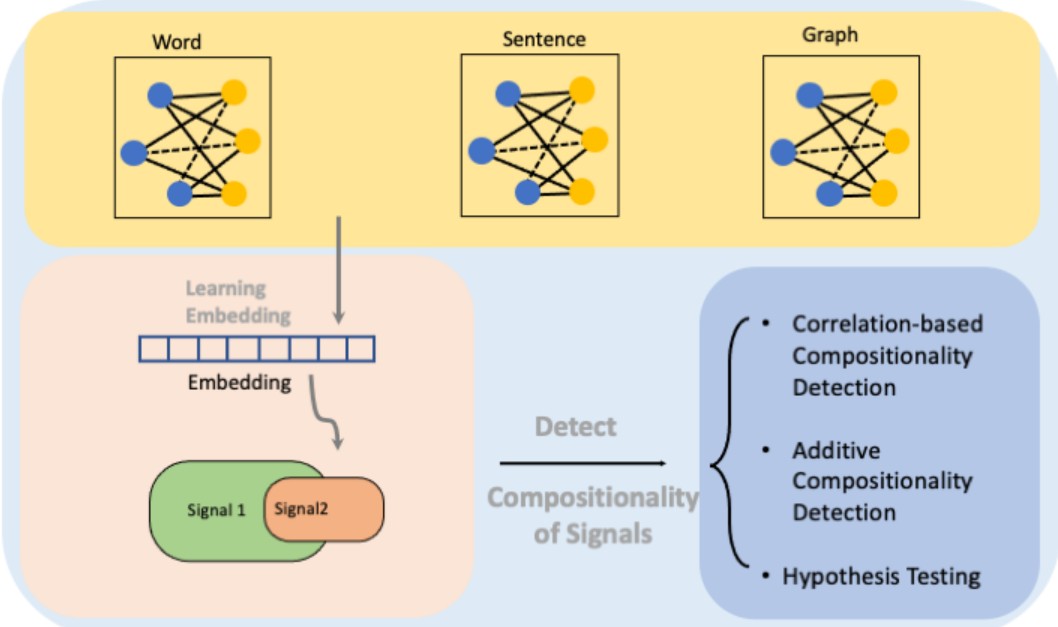

Figure 2: Structure of the paper

architectures: Continuous Bag-of-Words (CBOW) and Skip-Gram. CBOW predicts a word from its context, while Skip-Gram predicts context words from a target word.

## 2.2 Sentence Embedding: BERT

We also consider the problem of deriving the meaning of sentences from the meaning of the words within them. We look at sentence embeddings extracted from BERT. BERT, introduced by Devlin et al. (2018), is a pre-trained Transformer-based model capturing bidirectional contexts of words, producing nuanced sentence embeddings. SBERT (Reimers & Gurevych, 2019), a sentence embedding derivative of BERT, was trained on natural language inference (NLI) corpora (Bowman et al., 2015; Williams et al., 2018).

For each input token, BERT generates an output vector, where $\Phi_{BERT} : X \to Y \in \mathbb{R}^{768}$. The output vector of the [CLS] token is usually used for classification tasks because it can represent the information of the entire input sequence. However, the representation generated by pre-trained BERT fails to capture sentence similarity. Ideally, the sentence embeddings with similar meanings will be close to each other in the vector space. Thus, we use SBERT (Reimers & Gurevych, 2019), a version of BERT trained specifically for generating sentence representation that can be compared using cosine similarity. It created a leading performance on semantic textual similarity (STS) task (Cer et al., 2017) by introducing a Siamese structure. We use the CLS token from SBERT as our sentence embedding.

SBERT creates a state-of-the-art performance on variable STS tasks compared to previously existing sentence embeddings, such as InferSent (Conneau et al., 2017) and Universal Sentence Encoder (Cer et al., 2018). Using SBERT to generate sentence embedding helps us look into BERT's mechanism while investigating the compositionality in the embedding.

## 2.3 Knowledge Graph Embedding

A *graph* $G = (V, E)$ consists of a set of vertices $V$ with edges $E$ between pairs of vertices. In a *knowledge graph*, the vertices $V$ represent entities in the real world, and the edges $E$ encode that some relation holds between a pair of vertices. As a running example, we consider the case where the vertices $V$ are a set of viewers and films, and the edges $E$ encode the fact that a viewer has rated a film.

Knowledge Graphs represent information in terms of entities (or nodes) and the relationships (or edges) between them. The specific relation $r$ that exists between two entities is depicted as a directed edge, and this connection is represented by a triple $(h, r, t)$. In this structure, we distinguish between the two nodes involved: the *head* ($h$) and the *tail* ($t$), represented by vectors $\mathbf{h}$ and $\mathbf{t}$ respectively. Such a triple is termed a *fact*, denoted by $f$:

$$f = (h, r, t)$$

In order to mathematically capture the relationships and structures within a knowledge graph, we use the concept of embeddings. A knowledge graph embedding assigns vectors to nodes and edges in such a way that the graph's topology is encoded. To be specific, a vector $\mathbf{x} \in \mathbb{R}^n$ is allotted to each member of $V$, ensuring the existence of a distance function $D(\mathbf{x}_i, \mathbf{x}_j)$ where $E(v_i, v_j) = 1 \iff D(\mathbf{x}_i, \mathbf{x}_j) < \theta$ for a certain threshold $\theta$. We refer to these vectors $\mathbf{x}$ as the *embedding* of the nodes. The function that facilitates this embedding is the *embedding function*: $\Phi_{KG} : V \to \mathbb{R}^n$, or $\mathbf{x} = \Phi(v)$.

Conversely, given a set of points in a space, we can link them to form a graph. The decision of which pairs of nodes $\langle v_i, v_j \rangle$ should be linked is made by using a scoring function $f(\mathbf{x}_i, \mathbf{x}_j)$ that will be learnt from data. Two commonly used functions generating a score between $\mathbf{x}_i$ and $\mathbf{x}_j$ are:

$$\textbf{Multiplicative: } S(\mathbf{x_i}, \mathbf{x_j}) = \mathbf{x_i}^T \mathbf{R} \mathbf{x_j} \tag{1}$$

$$\textbf{Additive: } S(\mathbf{x_i}, \mathbf{x_j}) = \|\mathbf{x_i} + \mathbf{r} - \mathbf{x_j}\| \tag{2}$$

where $\mathbf{R}$ and $\mathbf{r}$ are parameterised matrices or vectors that will be defined below. We can think of different $\mathbf{R}_i$ and $\mathbf{r}_i$ as encoding specific relations, allowing the same entity embedding $\mathbf{x}$ to participate in multiple different relations.

We will follow this convention below, and use the multiplicative form of the scoring function which follows the settings of Berg et al. (2017)

**Multiplicative Scoring Function**   Nickel et al. (2011) proposed a tensor-factorisation based model for relational learning, in which they treat each frontal slice) of the tensor as a co-occurrence matrix for each entity with a given specific relation. Such a tensor could then be decomposed into three different tensors for the head entity, relation and tail entity. For example, consider a 3D tensor, and we are looking at its frontal slices. The $i, j$ entry of the $k$-th frontal slice encodes the interaction between the head entity $h_i$, the relation $R_k$, and the tail entity $t_j$. This entry can be decomposed into the product of $\mathbf{h_i}$, $\mathbf{R_k}$ and $\mathbf{t_j}$ A scoring function of a triple could also explain this in multiplicative way. We use $S(f)$ to denote the score of a triple $(h, r, t)$ and we use $\mathbf{h}, \mathbf{R}, \mathbf{t}$ (vectors) to denote the embeddings of each element of the triple $f = (h, r, t) \in F$.

$$S(f) = \mathbf{h}^T \mathbf{R} \mathbf{t} \qquad \mathbf{h} \in \mathbb{R}^d, \mathbf{R} \in \mathbb{R}^{d \times d}, \mathbf{t} \in \mathbb{R}^d \tag{3}$$

Various model variations exist. DistMult (Yang et al., 2014) retains only the $R$ matrix diagonal, reducing over-fitting. ComplEx (Trouillon et al., 2016) uses complex vectors for asymmetric relations.

In this work, we will be using DistMult (Yang et al., 2014) for the models. DistMult is favored for its simplicity and computational efficiency, especially its adeptness at capturing symmetric relations using element-wise multiplication of entity embeddings, which also makes it scalable for large knowledge graphs.

**Additive Scoring Function**   Bordes et al. (2013) introduced TransE, where relationships translate entities in the embedding space. For instance, $\mathbf{h}(King) + \mathbf{r}(FemaleOf) \approx \mathbf{t}(Queen)$.

$$S(f) = \|\mathbf{h} + \mathbf{r} - \mathbf{t}\| \qquad \mathbf{h} \in \mathbb{R}^d, \mathbf{r} \in \mathbb{R}^d, \mathbf{t} \in \mathbb{R}^d \tag{4}$$

**Rating Prediction**   In alignment with (Berg et al., 2017), we establish a function $P$ that, given a triple of embeddings $(\mathbf{h}, \mathbf{R}, \mathbf{t})$, calculates the probability of the relation against all potential alternatives.

$$P(\mathbf{h}, \mathbf{R}, \mathbf{t}) = \text{SoftArgmax}(S(f)) = \frac{e^{S(f)}}{e^{S(f)} + \sum_{r' \neq r \in \mathscr{R}} e^{S(f')}} \tag{5}$$

In the above formula, $f = (h, r, t)$ denotes a true triple, and $f' = (h, r', t)$ denotes a corrupted triple, that is a randomly generated one, that we use as a proxy for a negative example (a pair of nodes that are not connected).

Assigning numerical values to relations $r$, the predicted relation is then just the expected value prediction $= \sum_{r \in \mathscr{R}} r P(\mathbf{h}, \mathbf{R}, \mathbf{t})$ In our application of viewers and movies, the set of relations $\mathscr{R}$ could be the possible ratings that a user can give a movie. The predicted rating is then the expected value of the ratings, given the probability distribution produced by the scoring function. $S(f)$ refers to the scoring function in Yang et al. (2014).

To learn a graph embedding, we follow the setting of Bose & Hamilton (2019) as follows,

$$L = - \sum_{f \in \mathscr{F}} \log \frac{e^{S(f)}}{e^{S(f)} + \sum_{f' \in \mathscr{F}'} e^{S(f')}} \tag{6}$$

This loss function maximises the probabilities of true triples ($f$) and minimises the probability of triples with corrupted triples: ($f'$).

**Evaluation Metrics** We use 4 metrics to evaluate our performance on the link prediction task. These are root mean square error (RMSE, $\sqrt{\frac{1}{n} \sum_{i=1}^{n} (\hat{y}_i - y_i)^2}$, where $\hat{y}_i$ is our predicted relation and $y_i$ is the true relation), Hits@K - the probability that our target value is in the top $K$ predictions, mean rank (MR) - the average ranking of each prediction, and mean reciprocal rank (MRR) to evaluate our performance on the link prediction task. These are standard metrics in the knowledge graph embedding community.

## 3 Compositionality Detection Methods

An important consideration is that there is a difference between which information is *present* in a given data representation, and which information is *accessible* to a specific class of functions. While it may be difficult or impossible to prove that certain information is not present, it may be simple to prove that it is not accessible - say - to a linear function. In practical applications this may be all that is needed. For example, the study Jia et al. (2018) describes a method to ensure that a deep neural network does not contain unwanted information in a form that it can be used by its final - decision making - layers.

The general problem is as follows. Given a knowledge graph $G = (V, E)$, it may be the case that vertices $V$ have attributes that may be considered private information. For example, suppose we have a graph representing jobs and applicants. Suppose we have vertices representing applicants, vertices representing skills, and vertices representing jobs, with edges denoting which jobs applicants are finally offered. Some attributes of the applicants, for example their gender or age, may be considered *private information* that we do not wish to be able to elicit from the graph.

We give two methods: Correlation-based Compositionality Detection and Additive Compositionality Detection to detect the compositionality of signals in the vertices $V$. We take movie recommender system as a small running example.

### 3.1 Correlation-based Compositionality Detection

Canonical Correlation Analysis (CCA) is used to measure the correlation information between two multivariate random variables (Shawe-Taylor et al., 2004). Just like the univariate correlation coefficient, it is estimated on the basis of two aligned samples of observations.

A matrix of binary-valued attribute embeddings, denoted as $\mathbf{A}$, is essentially a matrix representation where each row corresponds to a specific attribute and each column corresponds to an individual data point (such as a word, image, or user). The entries of the matrix can take only two values, typically 0 or 1, signifying the absence or presence of a particular attribute. For example, in the context of textual data, an attribute might represent whether a word is a noun or not, and the matrix would be populated with 1s (presence) and 0s (absence) accordingly.

On the other hand, a matrix of user embeddings, denoted as $\mathbf{U}$, is a matrix where each row represents an individual user, and each column represents a certain feature or dimension of the embedding space. These embeddings are continuous-valued vectors that capture the movie preference of the users. The values in this matrix are not constrained to binary values and can span a continuous range.

Assuming we have a vector for an individual attribute embedding, denoted as

$$\mathbf{a} = (a_1, a_2, \ldots, a_n)^T$$

and a vector for an individual user embedding,

$$\mathbf{u} = (u_1, u_2, \ldots, u_m)^T$$

our goal is to explore the correlation between these two vectors. To achieve this, we focus on finding projection vectors, $\mathbf{w}_a$ (where $\mathbf{w}_{a_k} \in \mathbb{R}^n$) for the attribute and $\mathbf{w}_u$ (where $\mathbf{w}_{u_k} \in \mathbb{R}^m$) for the user, such that the correlation between the transformed embeddings is maximized. Mathematically, this can be expressed as:

$$\rho = \max_{\left(\mathbf{w}_{a_k}, \mathbf{w}_{u_k}\right)} \text{corr} \left(\mathbf{w}_{a_k}^T \mathbf{a}, \mathbf{w}_{u_k}^T \mathbf{u}\right) \tag{7}$$

Note there are $k$ correlations corresponding to $k$ components.

By extending the individual user case to all $q$ users, we can compute the canonical correlations for the entire user base, which provides insights into the relationship between the attribute embeddings and user embeddings across the whole dataset.

Given two matrices, one representing binary-valued attribute embeddings and the other representing user embeddings, we aim to find a correlation between them. Specifically, we define:

- $\mathbf{A}$: An $n \times q$ matrix of binary-valued attribute embeddings, where each column represents the attribute embeddings for a specific user, and $n$ is the number of attributes.

- $\mathbf{U}$: An $m \times q$ matrix of user embeddings, where each column represents the embedding of a different user, and $m$ is the dimensionality of each user embedding.

To compute the correlation between these matrices, we seek projection matrices $\mathbf{W}_A$ and $\mathbf{W}_U$ that maximize the correlation between the transformed $\mathbf{A}$ and $\mathbf{U}$. Formally, the objective is:

$$\rho = \max_{(\mathbf{W}_A, \mathbf{W}_U)} \text{corr} \left(\mathbf{A}\mathbf{W}_A, \mathbf{U}\mathbf{W}_U\right) \tag{8}$$

These paired random variables are often different descriptions of the same object, for example genetic and clinical information about a set of patients (Seoane et al., 2014), french and English translations of the same document (Vinokourov et al., 2002), and even two images of the same object from different angles (Guo & Wu, 2019).

In the example of viewers and movies, we use this method to compare two descriptions of users. One matrix is based on demographic information, which are indicated by Boolean vectors. The other matrix is based on their behaviour, which is computed by their movie ratings only.

**More specifically,** as shown in Figure 3, CCA seeks transformation vector $w_A(w_A \in R^{4 \times 1})$ and $w_B(w_B \in R^{3 \times 1})$ such that $\mathbf{w}_{a_k}^T \mathbf{a}$ and $\mathbf{w}_{u_k}^T \mathbf{u}$ maximize the correlation $\rho$ as shown in Equation 7.

For 4 different users, We define two transformation matrices $W_A(W_A \in R^{4 \times k})$ and $W_B(W_A \in R^{4 \times k})$, which stores k-pairs of transformation vector $w_A$ and $w_B$, as shown in Equation 8.

In the case of user embedding, CCA aims to learn the user attributes directions and user behavior directions in the embedding spaces so that the projections are maximally correlated.

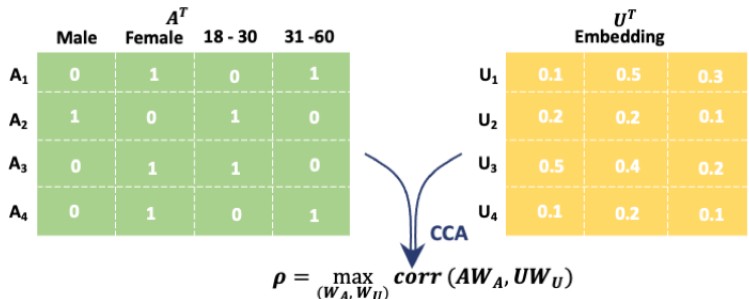

Figure 3: Schematic of Correlation-based compositionality Detection

## 3.2 Additive Compositionality Detection

Again assuming we have a matrix of entity embeddings $\mathbf{U}$ with matrix of attributes $\mathbf{A}$, we investigate the possibility that the entity embeddings can be decomposed into a linear combination of embeddings corresponding to attributes. Specifically, we investigate whether we can learn a matrix $\mathbf{X}$ as follows

$$\mathbf{AX} = \mathbf{U} \tag{9}$$

As mentioned in Section 2, word embeddings generated from the distribution of words in text can encode additional semantic or syntactic information. We investigate here the possibility that entity embeddings in knowledge graphs can be decomposed into linear combinations of embeddings corresponding to attributes. We use methods from Xu et al. (2023) to see if an entity embedding $\mathbf{u}$ can be decomposed into a linear system.

In our example of viewers and movies, a set of users as $U$ and the coefficient matrix of the components as $\mathbf{A}$. We aim to solve a linear system $\mathbf{AX} = \mathbf{U}$ so that the user embedding can be decomposed into three components (gender, age, occupation) as follows, $\mathbf{u} = \sum_i a_i \mathbf{x}_i$. Here, $\mathbf{u}$ is a user embedding, $i$ ranges over all possible values of each private attribute, $\mathbf{x}_i$ is an embedding corresponding to the $i$th attribute value, and $a_i \in \{0, 1\}$, denotes whether a particular attribute value is present or absent for the user. This formulation allows us to break down each user into distinct, quantifiable components, reflecting their demographics and interests.

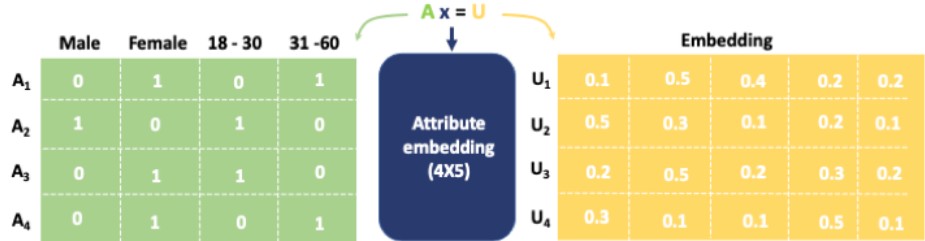

Figure 4: Schematic of Additive Compositionality Detection: our linear decomposition system

## 3.3 Hypothesis Testing with Random Permutations

We aim to investigate the correlation between user attributes and their movie preferences. By measuring a test statistic for correlation, and subsequently useing a permutation test on one of the datasets, we assess the likelihood of observing the same degree of correlation under the null hypothesis of no association.

To assess the significance of the observed correlation, a permutation test was conducted. This involved randomizing the order of users in one of the datasets (either attributes or movie preferences) while keeping the order in the other dataset unchanged. The test statistic for correlation was recalculated for each permutation.

Our null hypothesis is that the embedding of a vertex $u$ and its attributes $a$ are independent. To test whether this is the case, we use a non-parametric statistical test, whereby we directly estimate the $p$-value as the probability that we could obtain a "good"[1] value of the test statistic under the null hypothesis. If the probability of obtaining the observed value of the test statistic is less that $1\%$, we reject the null hypothesis.

Specifically, we will randomly shuffle the pairing of vertices and attributes 100 times, and compute the same test statistic. If the test statistic of the paired data is better than that of the randomly shuffled data across all 100 random permutations, we conclude that the correctly paired data performs better to a $1\%$ significance level.

The test statistic for *Correlation-based Compositionality Detection* is the correlation $\rho$. For the *Additive Compositionality Detection* $\mathbf{AX} = \mathbf{U}$, we use the Leave One Out algorithm as shown in Algorithm 1, that is to leave one user out and predict either the user embedding or the inverse problem of user identity. We look at the L2 norm loss of the linear system, cosine similarity and retrieval accuracy, a metric defined in Xu et al. (2023).

- L2 Loss of the linear system $||\mathbf{AX} - \mathbf{U}||^2$

- Cosine similarity between $\mathbf{u}$ and constructed embedding $\hat{\mathbf{u}}$

- Accuracy of retrieving identity of $\mathbf{u}$ with $\hat{\mathbf{u}}$

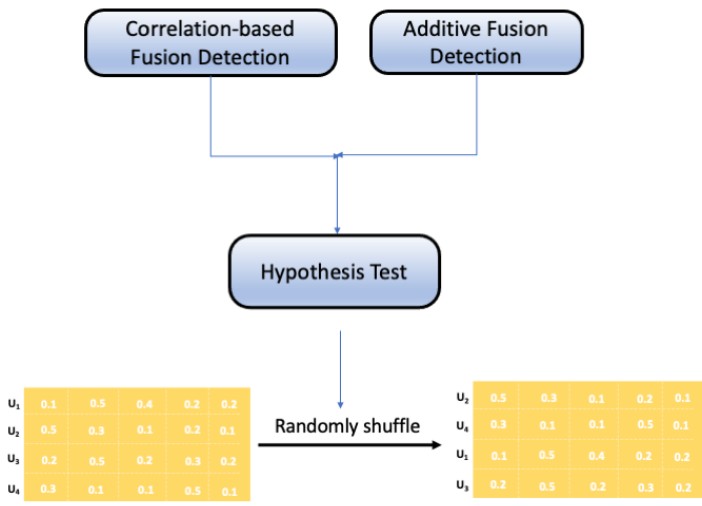

Figure 5: Hypothesis Testing

---

**Algorithm 1** Leave One Out

---

1: **for** any dataset of $(\mathbf{A}, \mathbf{U})$ descriptions **do**   ▷ (*)
2:     **for** each user $u$ **do**
3:         Leave the user $u$ out
4:         Train on the remaining $N - 1$ users
5:         Predict the user behavior $\hat{\mathbf{U}}$   ▷ (**)indicate the synthetic/predicted behavior with ^
6:         Measure the quality of $\hat{\mathbf{U}}$   ▷ (***)
7:     **end for**
8:     The Score is average quality (across all users) of artificial embeddings $\hat{\mathbf{U}}$
9: **end for**

---

[1] either high or low, depending on the statistic

---

**Algorithm 2** Compute a Loss Function

---

1: For a specific user, with true behavior $\mathbf{U}$ and predicted behavior $\hat{\mathbf{U}}$
- L2 Norm between $\mathbf{U}$ and $\hat{\mathbf{U}}$ : $||\mathbf{AX} - \mathbf{U}||^2$
- Cosine between $\mathbf{U}$ and $\hat{\mathbf{U}}$
- Identity between $\mathbf{U}$ and best_match_of: $\hat{\mathbf{U}}$

---

**Notes:**
(*) This includes randomly shuffled $(\mathbf{A}, \mathbf{U})$ pairs.
(**) Here, we take use as an example, the user behavior means user embedding computed by the movie preference, it could also be word/sentence embedding computed by the context.
(***) This includes different loss functions as shown in Algorithm 2.

**Hypothesis testing on Correlation-based Compositionality Detection**  In this study, we use a non-parametric testing approach to directly estimate the p-value as the probability of an event under the null hypothesis. This event pertains to the chance occurrence of a high value of the test statistic, specifically a strong correlation between two datasets. By leveraging a Monte Carlo sampling method, where random permutations of the user list serve as the basis for our samples, we assess the likelihood of observing the given test statistic purely by chance. If the probability of achieving the observed test statistic is less than 1%, we lean towards rejecting the null hypothesis. However, it is important to note that this does not conclusively affirm the alternative hypothesis ($H_1$) but rather emphasizes the statistical significance of our findings, a nuance that delves into the philosophical underpinnings of statistical inference.

**Hypothesis testing on Additive Compositionality Detection**  In this segment of the study, our objective is to substantiate the hypothesis that the embedding of user behaviour can be characterized by user demographics. We postulate that the representation of user behaviour, termed here as the "user-behaviour-embedding", can be approximated as a summation of vectors representing user demographics. To evaluate the accuracy of this approximation, we use a test statistic based on the loss or distance between the actual user behaviour embedding and its demographic-based approximation. A critical inquiry that emerges is: given the computed loss value, what is the probability that such a value could arise purely by chance under the null hypothesis? To address this, we implement a permutation-based approach, wherein we shuffle the data and estimate the probability of obtaining our observed test statistic under randomized conditions.

## 4  Experiments

We will examine the semantic and syntactic signals in word2vec embeddings, comparing them to WordNet and MorphoLex benchmarks. Subsequently, we will analyze the compositionality of BERT sentence embeddings, hypothesizing an additive relationship between individual word and complete sentence representations. Finally, using the MovieLens dataset, we will study the relationship between user movie preferences and demographic traits through behaviour-based embeddings.

### 4.1  Word Embedding

We are interested in examining two distinct signals encapsulated within the word2vec embeddings: semantic and syntactic information. To detect these signals, we use WordNet embeddings as semantic representation, and MorphoLex for syntactic structures. By comparing the word2vec embeddings against both WordNet and MorphoLex, we are able to disentangle and analyze the semantic and syntactic aspects of the word2vec representation.

#### 4.1.1  WordNet

WordNet (Miller, 1995) is a large lexical database of English, which consists of 40943 entities and 11 relations. WordNet is a combination of dictionary/thesaurus with a graph structure. Nouns, verbs, adjectives, and

adverbs are grouped into sets of cognitive synonyms (synsets), each expressing a distinct concept. These synsets are interlinked using conceptual-semantic and lexical relations.

The relations include, for instance, synonyms, antonyms, hypernyms (kind of relationship), hyponyms (part of relationship), meronyms (member of relationship), and more. For example, searching for 'ship' in WordNet might yield relationships to 'boat' (as a synonym), 'cruise' (as a verb related to 'ship'), or 'water' (as a related concept), among other things.

### 4.1.2 WordNet Embedding

We want to ensure our WordNet embedding can contain the semantic relation in it. Therefore, we train the embedding with the task of predicting the tail entity given a head entity and relation. For example, we might want to predict the hypernym of cat:

$$< \textbf{cat}, hypernym, \textbf{?} >$$

We train the WordNet Embedding in the following way:

1. We split our dataset to use 90% for training, 10% for testing.

2. Triples of $(head, relation, tail)$ are encoded as relational triples $(h, r, t)$.

3. We randomly initialize embeddings for each $h_i$, $r_j$, $t_k$, use the scoring function in Equation 4 and minimize the loss by Margin Loss.

4. We sampled 20 corrupted entities. Learning rate is set at 0.05 and training epoch at 300.

Results can be found in the Table 1, which shows that our WordNet embeddings do contain semantic information.

Table 1: Link prediction performance for WordNet

|  | Hits@1 | Hits@3 | Hits@10 | MRR |
|---|---|---|---|---|
| WordNet | 0.39 | 0.41 | 0.43 | 0.40 |

### 4.1.3 MorphoLex

MorphoLex(Sánchez-Gutiérrez et al., 2018) provides a standardized morphological database derived from the English Lexicon Project, encompassing 68,624 words with nine novel variables for roots and affixes. Through regression analysis on 4724 complex nouns, the dataset highlights the influence of root frequency, suffix length, and the prevalence of frequent words in a suffix's morphological family on lexical decision latencies. It offers valuable insights into morphology's role in visual word processing.

In this paper, we specifically focus on words with one root and multiple suffixes. For the CCA experiment, words with suffixes occurring less than 10 times are filtered out. In the linear decomposition experiment, we exclude rows with roots appearing fewer than 3 times.

### 4.1.4 Correlation-based Compositionality of Semantics and Morphology in Word2Vec

We applied Correlation-based Compositionality Detection to compare two different representations of a set of words. word2vec provides a vector space model that represents words in a high-dimensional space, using the context in which words appear.

Table 2: Suffix presence (indicated by '1') for selected words from the MorphoLex dataset

| Word | al | ic | ist | ity | ly | y |
|---|---|---|---|---|---|---|
| allegorically | 1 | 1 | 0 | 0 | 1 | 0 |
| whimsicalities | 1 | 0 | 0 | 1 | 0 | 1 |
| whimsicality | 1 | 0 | 0 | 1 | 0 | 1 |
| whimsically | 1 | 0 | 0 | 0 | 1 | 1 |
| voyeuristically | 0 | 1 | 1 | 0 | 1 | 0 |

**Semantics** WordNet offers a structured lexical and semantic resource where words are related based on their meanings and are organized into synonym sets. We shuffled the pairing of word2vec embedding and words 100 times to break the semantic signal captured in the word2vec embedding. The result is shown in Figure 6a. The correlation between two different representations is higher than the shuffled ones in the first component, which means that information about the semantic relations between words can be captured from the word embedding trained by its context words.

**Morphology** MorphoLex provides a morphological resource predicated on root frequency, suffix length, and the function of morphology. We permuted the word2vec embedding 50 times to hide the morphological signals in the word2vec representation, with results reported in Figure 6b. We see that the correlation coefficient observed between the word2vec and MorphoLex embeddings is significantly higher than the random baseline. This suggests that morphology is represented in word2vec embeddings for multiple suffixes at a time.

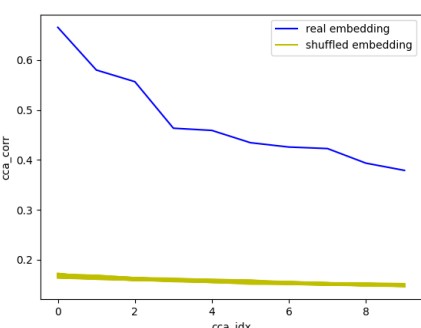

(a) PCC for the true WordNet-word2vec pairings and 100 permuted pairings. The first 10 components are selected for illustration. WordNet embeddings contain semantic information.

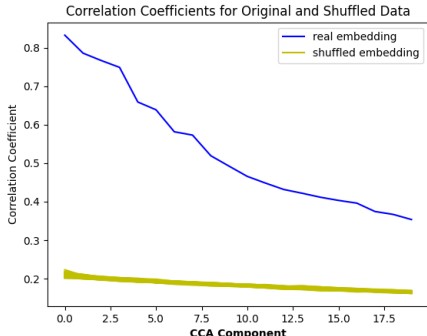

(b) PCC comparison for the true MorphoLex-word2vec pairings and 100 permuted pairings. The first 20 components are selected for illustration. MorphoLex embeddings contain the morphological information.

Figure 6: Comparison of PCC values for WordNet-word2vec and MorphoLex-word2vec pairings. PCC is calculated between projected $\mathbf{A}$ and projected $\mathbf{U}$. $x$ axis stands for the $k$th components, $y$ axis gives the value. The PCC value for real pairings is larger than for any permuted pairings, which means both semantic and morphological information can be detected from the word2vec embedding.

### 4.1.5 Decomposing Word2Vec Embedding by Additive Compositionality Detection

We have chosen a collection of 278 words, where several words have common roots, and others have identical morphological units. Having computed a set $\mathbf{U} \in \mathbb{R}^{278 \times 300}$ of embeddings as word2vec embeddings, we can find the unknown vectors $\mathbf{x}_i$, $\mathbf{x}_j$, and $\mathbf{x}_k$ by solving the linear system $\mathbf{AX} = \mathbf{U}$, where $\mathbf{A} \in \mathbb{R}^{278 \times 45}$ is a binary matrix indicating the presence or absence of each root words and morphemes. This system does not

have (in general) an exact solution, so we approximate the solution by solving a linear least squares problem, using the Moore-Penrose pseudo-inverse.

We use a leave-one-out approach, training the linear system without including a target word $u$. We test the accuracy of this method by estimating the embedding for $u$ and comparing it to its true word2vec embedding, using the evaluation steps outlined in Algorithm 2.

Results are presented in Figure 7. We see that word2vec embeddings can be decomposed into root and multiple suffixes fairly well. The linear system loss is 39.84, lower than the minimum loss of the random system (44.06). Cosine similarity is 0.44, greater than all instances of the random baseline, and retrieval accuracy @ 10 is greater than that of the random system. However, overall these values are low, showing that there is still a fair bit of information that is not being captured by this representation.

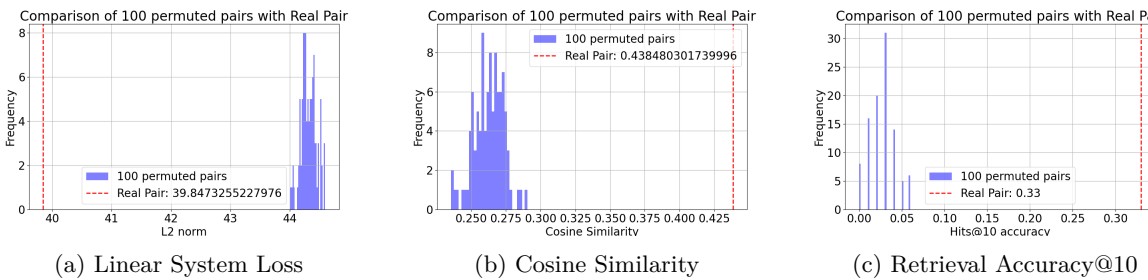

| (a) Linear System Loss | (b) Cosine Similarity | (c) Retrieval Accuracy@10 |

Figure 7: The test statistics for word2vec embedding decomposition. Dash line is the average performance of $\hat{\mathbf{B}}$ learned from the word2vec embedding. The bars are the distribution of the results from random permutations that run for 100 times.

## 4.2 Sentence Embedding

Following the decomposition of word2vec embeddings, we are further interested in whether sentence embeddings can be decomposed in a similar way. Sentences are compositional structures that are built from words. Therefore, it is natural to ask if the learned representations reflect the compositionality. We concentrate on BERT embedding models and investigate compositionality in sentence embeddings over layers and over training stages. We assume that there is an additive compositionality between words and sentences so that the sentence representation can be decomposed as

$$\Phi_{BERT}(Sentence) \approx \Phi(Word_1) + \cdots + \Phi(Word_N)$$

To investigate compositionality in BERT sentence embedding, we generated a sentence corpus that includes 1,000 sentences of the form subject-verb-object. We chose this sentence form in order to be able to easily analyse how different word types contribute to the overall sentence embedding.

### 4.2.1 Data Generation

We constructed a sentence corpus with 30 distinct components categorized into subjects ($Sbj$), verbs ($Verb$), and objects ($Obj$), which we then arranged into 10x10x10 triplet combinations of ($Sbj, Verb, Obj$). These triplets form short sentences using consistent prepositions and articles. For instance, the triplet ($cat, sat, mat$) yields the sentence "The cat sat on the mat." Our corpus comprises 1000 such sentences, enabling analysis of each component's role when decomposing with a linear system.

BERT uses a subword tokenization strategy, splitting words like "bookshelf" into "book" and "shelf". We selected corpus words to maintain uniform token counts across sentences. Since BERT considers punctuation as tokens, each sentence amounts to seven tokens.

To construct a sentence (I), we concatenate subject, verb, and object with indices $i$, $j$, and $k$ respectively. Thus, $I_{ijk} = Sbj_i \ Verb_j \ Obj_k$. We calculate sentence embedding $\mathbf{U}_{ijk} = \Phi_{BERT}(I_{ijk})$ with a pre-trained SBERT introduced in section 2.2.

We used the BERT-base SBERT model, which consists of 12 layers, to derive representations from various layers. Additionally, we used MultiBERTs (Sellam et al., 2022) to access intermediate checkpoints recorded during the pretraining steps of BERT.

### 4.2.2 Decomposing Sentence BERT Embedding by Additive Compositionality Detection

Given a set of sentence embeddings $\mathbf{U}$, we determine the unknown vectors $\mathbf{x}_i$, $\mathbf{x}_j$, and $\mathbf{x}_k$ by resolving $\mathbf{AX} = \mathbf{U}$. We term the embedding of a word $w$ obtained by this method $\Phi_C(w)$. Here, $\mathbf{A}$ is a $1000 \times 30$ binary matrix specifying each sentence component, $\mathbf{X}$ represents the $30 \times 768$ BERT embeddings for sentence attributes, and $\mathbf{U}$ is the $1000 \times 768$ matrix of sentence embeddings. The solution is obtained via the pseudo-inverse method. The embedding accuracy is quantified by the loss $L$, defined as:

$$L = \|\mathbf{AX} - \mathbf{U}\|^2 \tag{10}$$

For our null hypothesis, sentence embeddings are randomized to disrupt the sentence-embedding association, and loss is computed for this perturbed data over 100 iterations.

One of the challenges is if we can predict the sentence embedding $\mathbf{u}$ with the word representations solved by the linear system without seeing the actual sentences. To test this, we use the leave-one-out strategy, excluding the target sentence $I$ from the dataset while training the linear system. and reconstruct the sentence embedding by adding up the word representations we obtained so that

$$\Phi_C(I) = \Phi_C(Sbj) + \Phi_C(Verb) + \Phi_C(Obj) \tag{11}$$

We assess the embeddings obtained via two methods: first, by calculating the cosine similarity between the predicted and real embeddings; second, by determining if the predicted embedding can identify the correct sentence among 1000 possibilities.

### 4.2.3 Results

Figure 8 illustrates the performance of decomposing BERT sentence embedding. These results show that the BERT sentence embedding can be decomposed into three separate components: subject, verb, and object. Those components can then be used to predict the embedding of a new sentence.

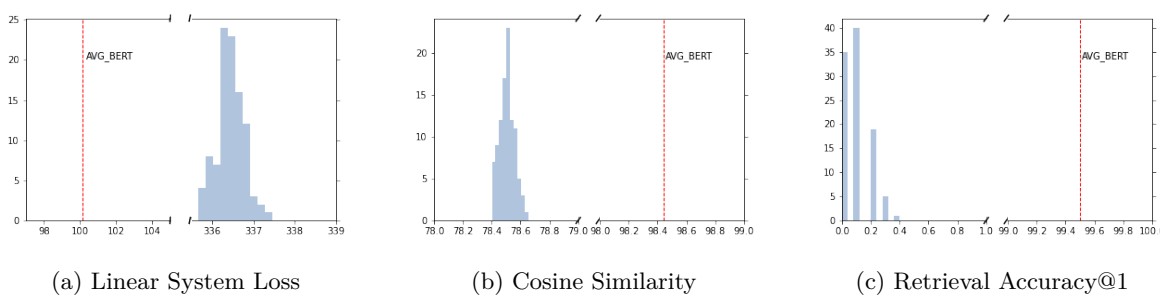

(a) Linear System Loss      (b) Cosine Similarity      (c) Retrieval Accuracy@1

Figure 8: The test statistics for sentence embedding decomposition. AVG_BERT is the average performance of $\hat{\mathbf{B}}$ learned from the BERT embedding. The bars are the distribution of the results from random permutations that run for 100 times (Xu et al., 2023).

The sentence embedding decomposition via the linear system yields a loss of 100.14, significantly less than the smallest loss from random permutations at 335.65 (significance level $\alpha = 0.01$). We note that the loss of 100.14 is across all 1000 embeddings, implying that the average Euclidean distance between the embedding obtained from SBERT and the reconstructed embedding is approximately 0.1. We conclude that the SBERT sentence embeddings are effectively representable by the sum of their Sbj, Verb, and Obj components. This conclusion is supported by the observation that the average cosine similarity between the SBERT sentence embedding and the reconstructed embedding is 0.98. Furthermore, $\hat{\mathbf{U}}$ achieves a 99.5%

success rate in retrieving the correct BERT embedding, whereas the best retrieval accuracy using randomized attribute/embedding pairings does not exceed 0.4%.

While these results are encouraging, it is the case that the other tokens in the sentence are created in the same way as the [CLS] token used for the sentence embedding. We carry out the same experiment across 30 random seeds using each of these other tokens as the representation of the sentence. This forms a more challenging baseline than a random permutation of embedding pairings. In this experiment, we replace some multi-token words with single token words. Specifically, we change "hamster" to "bear", "hedgehog" to "fox", "bookshelf" to "book".

Table 3 shows the performance of the linear system for each token level. We used three metrics to assess compositionality: L2 loss, cosine similarity, and retrieval accuracy.

We see that across all metrics, there are no substantial differences in whether the candidate sentence embedding can be decomposed into subject, verb, and object. The similarity of each token's embedding in the last layer of the transformer suggests a lack of distinct information among them, a finding that is supported by Park & Kim (2021). However, under the key performance metric of retrieval accuracy, the CLS token embedding does perform best. We performed a t-test and found that the retrieval accuracy of the embeddings produced from the CLS token is significantly higher ($p < 10^{-4}$) than the retrieval accuracies of the other tokens.

Table 3: Compositionality for each word in the sentence. Note that in this experiment, we replace some multi-token words into single token words. Specifically, we change "hamster" to "bear", "hedgehog" to "fox", "bookshelf" to "book".

| Metric | Values | | |
|---|---|---|---|
| | L2 Loss | Cosine Similarity | Retrieval Accuracy |
| **CLS** | 103.91 | 0.983 | 0.995 |
| **Subject** | 96.46 | 0.985 | 0.988 |
| **Verb** | 108.58 | 0.980 | 0.993 |
| **Object** | 102.55 | 0.983 | 0.991 |
| **The first 'The'** | 101.01 | - | - |
| **The last '.'** | 97.13 | 0.984 | 0.992 |
| **Random Baseline of CLS** | 343 | 0.77 | 0.01 |

**Compositionality across layers of SBERT**  We further investigate the differences between token embeddings through the layers of SBERT. We repeat the same experiment for each token and for each layer of the model. Table **??** reports the metrics for the CLS token, and Figures 9, 10, and 11 show the L2 loss, cosine similarity and retrieval accuracy across the 12 layers of SBERT. More detailed results correspongin to these figures are shown in appendix Table 7 and Table 8.

We see that through early layers of SBERT, up to layer 9, the CLS token is more amenable to decomposition into component word embeddings than the other tokens are, with lower loss, higher cosine similarity, and higher retrieval accuracy than the other token embeddings. We further see that in the earlier layers of the model, the CLS token exhibits more additive compositionality than in later layers. This indicates that as the sentence is processed through the layers, more contextual information is being added. However, what is interesting is that the amount of contextual information being added is still low, and much of the embedding can be accounted for additively. Considering cosine similarity (Figure 10), we see that the cosine similarity of the CLS embedding with its reconstruction drops from 1 in very early layers to around 0.98, indicating that a large proportion of the composition in the sentence can be interpreted additively.

Each transformer layer in SBERT consists of two main components: a multi-head self-attention mechanism and a position-wise feed-forward network. This contributes to refining the representations, making them richer and more context-aware. We are interested in measuring the compositionality in these represenatations

of different layers. Figure 9, 10, 11show the L2 loss, cosine similarity and retrieval accuracy across 12 layers of SBERT Embedding.

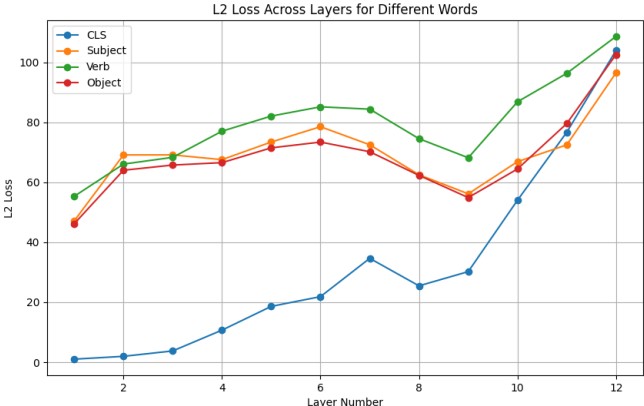

Figure 9: L2 Loss Across Layers in SBERT Embedding

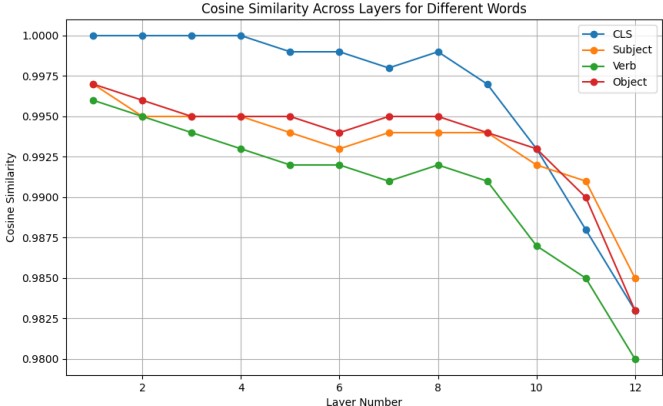

Figure 10: Cosine Similarity Across Layers in SBERT Embedding

**Compositionality across BERT training stages.** We also looked into how compositionality is captured during different training stages of BERT. We used the MultiBERTs Sellam et al. (2022) to get intermediate checkpoints captured during pre-training steps. Results are shown in Table 5 and Figure 12.

We see that at the beginning of training, the cosine similarity between the CLS embedding and the reconstructed embedding is very high, with perfect retrieval accuracy. We conjecture that this is due to the initialization of the BERT model. Sellam et al. (2022) use a GELU activation function and initialize the model with weights drawn from a truncated normal distribution with mean 0 and standard deviation 0.02. Close to 0, the GELU activation function is approximately linear. This means that the CLS token will naturally decompose into a weighted sum of component vectors.

As training progresses, we see that cosine similarity between the CLS token and its reconstruction decreases, meaning that more contextual information is added. However, what is interesting is that this again does not decrease by a large amount - the main component of the composition in these sentences is still additive.

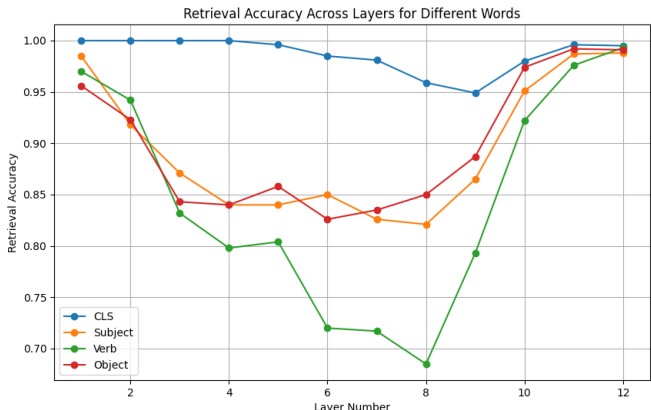

Figure 11: Retrieval Accuracy Across Layers in SBERT Embedding

Table 4: Compositionality for CLS Token in a Sentence (layer 1 to layer 12)

| Metric | Values | | |
|---|---|---|---|
| | L2 Loss | Cosine Similarity | Retrieval Accuracy |
| Layer 1 | 1.05 | 1.0 | 1.0 |
| Layer 2 | 1.99 | 1.0 | 1.0 |
| Layer 3 | 3.79 | 1.0 | 1.0 |
| Layer 4 | 10.72 | 1.0 | 1.0 |
| Layer 5 | 18.64 | 0.999 | 0.996 |
| Layer 6 | 21.88 | 0.999 | 0.985 |
| Layer 7 | 34.64 | 0.998 | 0.981 |
| Layer 8 | 25.48 | 0.999 | 0.959 |
| Layer 9 | 30.22 | 0.997 | 0.949 |
| Layer 10 | 54.01 | 0.993 | 0.98 |
| Layer 11 | 76.66 | 0.988 | 0.996 |
| Layer 12 | 103.91 | 0.983 | 0.995 |

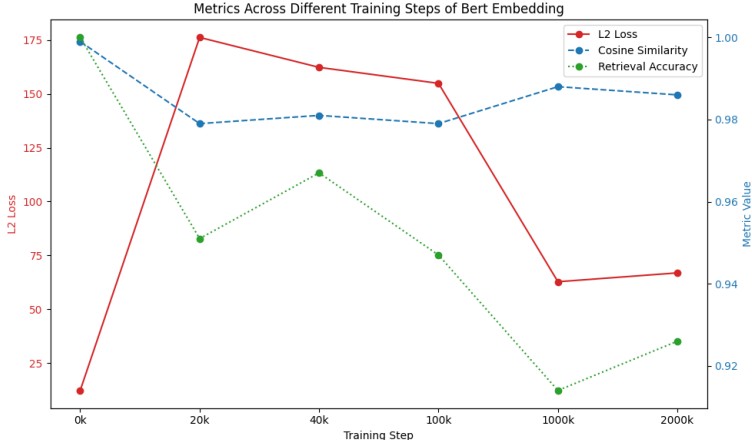

Figure 12: Metrics Across Different Training Steps of Bert Embedding

Table 5: Compositionality in Different Training Steps of Bert Embedding

| Training Step | Values | | |
|---|---|---|---|
| | L2 Loss | Cosine Similarity | Retrieval Accuracy |
| **CLS__0k** | 12.23 | 0.999 | 1.0 |
| **CLS__20k** | 176.24 | 0.979 | 0.951 |
| **CLS__40k** | 162.33 | 0.981 | 0.967 |
| **CLS__100k** | 154.90 | 0.979 | 0.947 |
| **CLS__1000k** | 62.74 | 0.988 | 0.914 |
| **CLS__2000k** | 66.85 | 0.986 | 0.926 |
| **Random Baseline of CLS__2000k** | 141 | 0.93 | 0.01 |

### 4.3 Knowledge Graph Embedding

As described in section 2.3, knowledge graphs can be represented as sets of node embeddings and relation embeddings that reflect the structure of the graph, given a scoring function. In these experiments, we examine the extent to which *attributes* of nodes can be predicted from embeddings, even when those attributes are not used in the training of the node embeddings. We train our model on GeForce GTX TITAN X.

### 4.3.1 Datasets

This experiment was conducted on the MovieLens 1M dataset (Harper & Konstan, 2015) which consists of a large set of movies and users, and a set of movie ratings for each individual user. It is widely used to create and test recommender systems. Typically, the goal of a recommender system is to predict the rating of an unrated movie for a given user, based on the rest of the data. The dataset contains 6040 users and approximately 3900 movies. Each user-movie rating can take values in 1 to 5. There are 1 million triples (out of a possible $6040 \times 3900 = 23.6m$), so that the vast majority of user-movie pairs are not rated.

Users and movies each have additional attributes attached. For example, users have demographic information such as gender, age, or occupation. Whilst this information is typically used to improve the accuracy of recommendations, we use it to test whether the embedding of a user correlates to private attributes, such as gender or age. We compute our graph embedding based only on ratings, leaving user attributes out. Experiments for training knowledge graph embeddings are implemented with the OpenKE (Han et al., 2018) toolkit.

We embed the knowledge graph in the following way:

1. We split our dataset to use 90% for training, 10% for testing.

2. Triples of $(user, rating, movie)$ are encoded as relational triples $(h, r, t)$.

3. We randomly initialize embeddings for each $h_i$, $r_j$, $t_k$ and train embeddings to minimize the loss in equation 6.

4. We sampled 10 corrupted entities and 4 corrupted relations per true triple. Learning rate is set at 0.01 and training epoch at 300.

We verify the quality of the embeddings by carrying out a link prediction task on the remaining 10% test set. We achieved a RMSE score of 0.88, Hits@1 score of 0.46 and Hits@3 as 0.92, MRR as 0.68 and MR as 1.89.

We now apply our three methods for bias detection to investigate the extent to which private information can be detected in node embeddings.

### 4.3.2 Correlation-based Compositionality Detection

We collect attribute information for all 6040 users and embed their personal attributes with Boolean indicator vectors $\mathbf{a}_i$ which encode the value of each attribute (gender, age, and occupation). We investigate whether users' private traits may be leaked from the graph embeddings by comparing two different user representations $\mathbf{a}_i$, the Boolean vector of attributes, and $\mathbf{u}_i$, the user embedding calculated as in section 4.3.1.

We apply CCA to calculate the correlation between users and their attributes. We apply the non-parametric statistical test described in section 3.3. Specifically, our null hypothesis is that users' movie preferences are not correlated with their attributes. We calculate Pearson's correlation coefficient (PCC) between projected $\mathbf{A}\mathbf{w}_A$ and projected $\mathbf{U}\mathbf{w}_U$. We go on to calculate the PCC between 100 randomly generated pairings of user and attribute embeddings, and find that the PCC between true pairs of attribute and user embeddings is higher each time. We therefore reject the null hypothesis at a 1% significance level. The correlation coefficients between real pairs and random pairs is reported in figure 13a.

Figure 13b displays weights indicating the contribution of each component to the overall attribute compositionality as determined by the correlation-based compositionality detection.

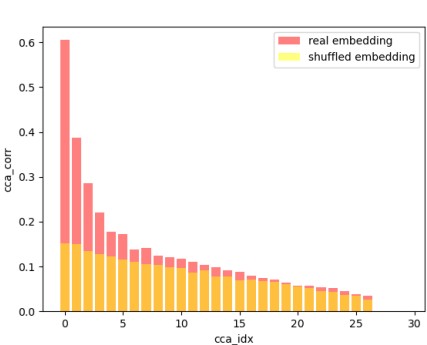

(a) Pearson's correlation coefficient (PCC) for true user-attribute pairings and 100 permuted pairings. PCC is calculated between projected $\mathbf{A}$ and projected $\mathbf{U}$. $x$ axis stands for the $k$th components, $y$ axis gives the value. The PCC value for real pairings is larger than for any permuted pairings.

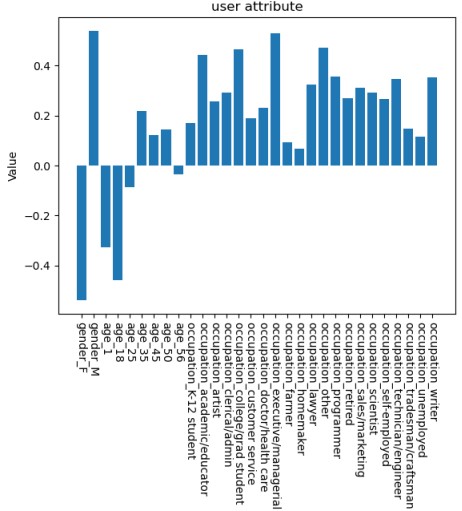

(b) Distribution for each attribute on the second component of CCA

Figure 13: Comparative analysis of PCC values and attribute distribution in CCA components.

We also carried out experiments on comparing user embedding of different stages. As shown in Figure 14, more demographic information of a user is encoded with more steps of training.

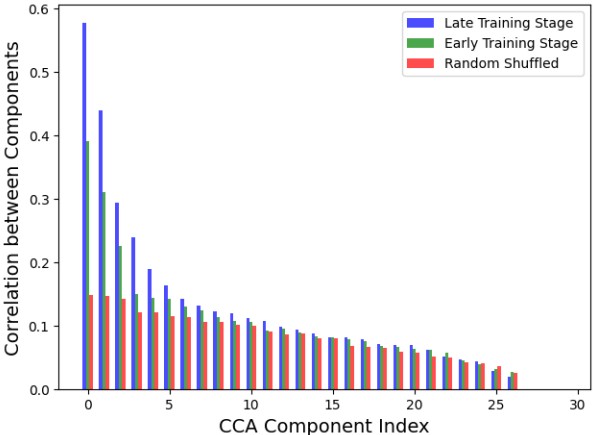

Figure 14: Demographics Encoded in Different Training Stages of Knowledge Graph Embedding

### 4.3.3 Additive Compositionality Detection on Gender and age

Preliminary results indicated a certain level of correlation between user attributes and movie preferences as measured by the test statistic. Subsequent permutation tests revealed that the observed correlation was rarely, if ever, achieved under randomized conditions.

We investigate the ability of a user embedding to be reconstructed as a linear sum of attribute embeddings by doing the leave-one-out experiment. We then try to interpret the knowledge graph embedding with user attributes. Similar to sentence embedding, a linear system is used to calculate the representation for each user attribute. Note that not all of the combinations of attributes exist in the movie lens dataset. We find that a user embedding can be reconstructed as a linear combination of its attributes by solving the linear system described in section 3.2. We use the pseudo-inverse method to solve this system. We try to interpret the user embedding with user attributes such as gender and age. we first group the user by age and gender firstly and compute the mean embedding of 14 group of users (2 gender groups and 7 age groups, giving 14 different gender-age combinations in total) and we take the mean embedding of each group. We use three test statistics as mentioned in Section 3.3 to test our linear system. We set a significance threshold: $\alpha = 0.01$.

Same as the Correlation-based Compositionality Detection setting, we permuted the pairing of users 100 times. Table 6 shows the observed p-value for three different statistics, which is the probability of seeing that value of statistic under the null hypothesis. We first decompose the user embedding into gender and age. Our results show the linear system is able to decompose the user embedding with a loss of 0.42 which is lower than every loss for a random permutation (1.03-1.96). The cosine similarity is 99.8%, higher than any permuted pairs. The identity retrieval accuracy is 0.93 which is higher than any random permuted pairs (0.0-0.21). Therefore, the null hypothesis is rejected. This shows that a user embedding can be reconstructed as a linear combination of gender and age.

This shows that attribute (demographic) embeddings can be learnt from user embeddings (behaviours), even when those attributes are not used in training the user embeddings, and moreover, the user embeddings are well represented as a sum of those embeddings.

The comparison with the randomly permuted embedding: beating the baseline means that we do detect a signal that needs explaining: that addition in the embedding space is what encodes the composition of meaning.

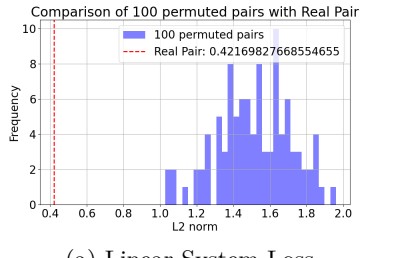 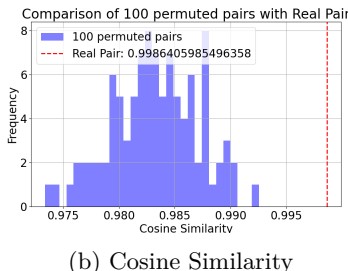 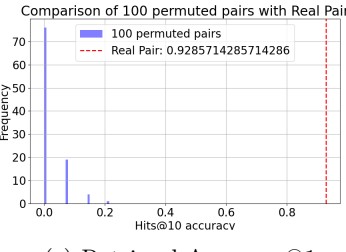

(a) Linear System Loss          (b) Cosine Similarity          (c) Retrieval Accuracy@1

Figure 15: The test statistics for user embedding decomposition. Dash line is the average performance of $\hat{\mathbf{B}}$ learned from the user embedding. The bars are the distribution of the results from random permutations that run for 100 times.

Table 6: p-value for hypothesis test. Note that * indicates better than random baseline to significance level $\alpha = 0.01$. In our case, we are estimating directly the p-value, as the probability of an event, that we could have a high (low) value of the test-statistic by chance under the null-hypothesis

|                              | L2 Norm       | Cosine Similarity | Retrieval Acc. | p-value  |
|------------------------------|---------------|-------------------|----------------|----------|
| Gender, Age Real Pair        | 0.42*         | 99.8%             | 0.93*          | <0.01    |
| Gender, Age Permuted         | 1.03-1.96*    | 97.3%-99.2%       | 0.00-0.21*     | <0.01    |
| Gender, Age, Occ Real Pair   | 17.54*        | 97.6%             | 0.23*          | <0.01    |
| Gender, Age, Occ Permuted    | 18.42-19.13*  | 96.8%-97.3%       | 0.00-0.07*     | <0.01    |

### 4.3.4 Additive Compositionality Detection on Gender, Age and Occupation

We afterwards group the user by gender, age and occupation and compute the mean embedding of 241 group of users. The number 241 is derived as follows. There are 294 potential combinations of 2 gender groups, 7 gender groups and 21 occupation groups. Only 241 of these gender-age-occupation combinations are actually present in the dataset.

When decomposing the embedding into gender, age and occupation, the L2 norm is 17.54 which is lower than every loss for a random permutation (18.42-19.13). As for identity retrieval accuracy, although the value is only 0.23 which is not a good result, it is still higher than any random permuted pairs (0.00-0.07). Therefore, the null hypothesis is rejected. Detailed information is shown in Figure 16.

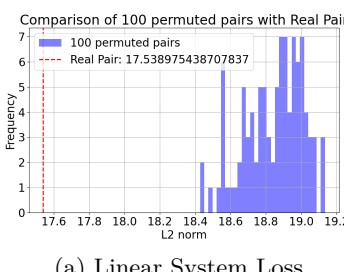 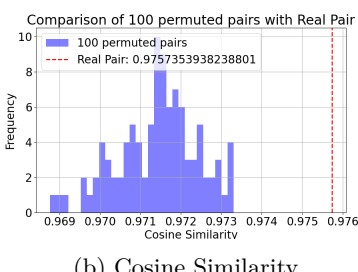 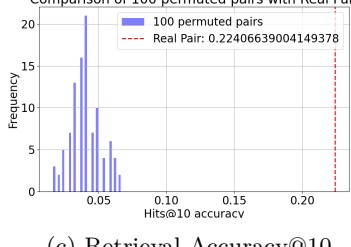

(a) Linear System Loss          (b) Cosine Similarity          (c) Retrieval Accuracy@10

Figure 16: The test statistics for user embedding decomposition. Dash line is the average performance of $\hat{\mathbf{B}}$ learned from the user embedding. The bars are the distribution of the results from random permutations that run for 100 times.

# 5    Discussion

We have presented two methods for signals of compositionality detection in three different data types, word embedding, sentence embedding and graph embedding.

**Word Embedding**   We showed that word2vec is able to capture semantic relationships like hyponymy to some extent. Even though word2vec operates in a continuous vector space, it surprisingly aligns well with semantically organized databases such as WordNet. Further, when analyzed alongside tools like MorphoLex, we see that word2vec embeddings can also be partially decomposed into roots and multiple suffixes. These observations emphasize the depth of information embedded within word contexts — they don't just convey basic meaning, but also carry detailed linguistic information, including morphology. Since word2vec captures some of both semantics and morphology, it may be the case that the weaker performance on each dataset is a result of having to encode both of these aspects of word meaning. Future work will look into developing a dataset that includes both semantic and morphological information in one setting, and use this to assess the additive compositionality of various embedding types.

**Sentence Embedding**   To examine the properties of sentence embedding, we generated an SVO sentence corpus and embedded it with SBERT. By applying a linear system, we showed that the SBERT sentence embedding can be decomposed into word representation with a linear system so that $\Phi_{SBERT}(I_{ijk}) \approx \Phi_{LINEAR}(Sbj_i) + \Phi_{LINEAR}(Verb_j) + \Phi_{LINEAR}(Obj_k)$. This allows for inference of a sentence embedding with simple linear algebra. The reconstructed embedding has average 0.98 cosine similarity with the CLS embedding, whereas the random baseline has a lower similarity of 0.77. The results show that the SBERT sentence embedding can be decomposed into a sum of indivudual word. However, it also contains some contextual information. Analysis of this contextual information remains future work. Although the SVO sentence format is simple, it is surprising that an SVO sentence can be decomposed into a sum of word embeddings, given that the processing of the words involves multiple nonlinearities.

We further found that other token embeddings within a sentence at the last layer in SBERT are able to be decomposed into component words, but that the CLS token is strongest when looking at the key metric of retrieval accuracy. Further, this effect is heightened at earlier layers of the model. We further found that the additive compositionality becomes weker through training: at the start of training, the CLS token can be perfectly decomposed into component word embeddings, whereas later in training, this is lessened. However, this is really only lessened very slightly: retrieval accuracy reduces from 1 to 0.926. This is indicative that additive compositionality does play a strong role in how BERT and potentially other Transformer models carry out composition, and can potentially contribute to explainability of these models. We will examine explainability in future work, as well examining whether the results hold for longer sentences or more complicated forms of meaning composition.

**Graph Embedding**   we found that certain dimensions of user embeddings that relate to specific information should correlate with certain patterns of demographic information corresponding to the same meaning, across all users. Using the private attributes representation obtained in this way we first demonstrate that the correlations detected between the two versions of the user representation are significantly higher than random, and hence that a representation based on such features does capture statistical patterns that reflect private attribute information.

As for the linear system, we assume that user-behaviour-embedding is (approximated by) a sum of user-demographic vectors, showing that user embeddings can be decomposed into a weighted sum of attribute embeddings. This refers to the compositionality of the user embedding, for example, the embedding of a "50 year old female" can be computed by the sum of the embedding of "50" and "female". Unlike the BERT models, where additive compositionality decreases through training, we saw that additive compositionality increases as we train the embeddings for longer. Of course, the model is very different (and more similar to the word2vec architecture). Further investigation of the differences in additive compositionality between these kinds of model is an area of future work

# 6 Conclusions

Three different types of data, word embedding, sentence embedding and knowledge graph embedding, present some compositionality, that is some of the information contained in them can be explained in terms of known attributes. This creates the possibility to manipulate those representations, for the purpose of removing bias, or to explain the decisions of the algorithm using them, or to answer analogical or counterfactual questions.

In the case of word embedding, both the semantic and morphological information signals are detected from the context-based embedding. Sentence embedding, produced by BERT, presents some compositionality in terms of subject, verb, and object. In the case of movie recommender system, computed by the movie preference only, user embedding presents some compositionality of their private attributes such as age, gender and occupation. This creates the possibility to manipulate those representations, for the purpose of removing bias, or to explain the decisions of the algorithm using them, or to answer analogical or counterfactual questions.

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

# A    Appendix

Table 7: Compositionality for Different Words in a Sentence (layer 1 to layer 6)

| Metric | Values | | |
|---|---|---|---|
| | **L2 Loss** | **Cosine Similarity** | **Retrieval Accuracy** |
| | **Initial Embedding** | | |
| | **Layer 1** | | |
| **CLS** | 1.05 | 1.0 | 1.0 |
| **Subject** | 47.13 | 0.997 | 0.985 |
| **Verb** | 55.31 | 0.996 | 0.97 |
| **Object** | 46.18 | 0.997 | 0.956 |
| | **Layer 2** | | |
| **CLS** | 1.99 | 1.0 | 1.0 |
| **Subject** | 69.10 | 0.995 | 0.918 |
| **Verb** | 66.05 | 0.995 | 0.942 |
| **Object** | 64.00 | 0.996 | 0.923 |
| | **Layer 3** | | |
| **CLS** | 3.79 | 1.0 | 1.0 |
| **Subject** | 69.11 | 0.995 | 0.871 |
| **Verb** | 68.30 | 0.994 | 0.832 |
| **Object** | 65.73 | 0.995 | 0.843 |
| | **Layer 4** | | |
| **CLS** | 10.72 | 1.0 | 1.0 |
| **Subject** | 67.53 | 0.995 | 0.84 |
| **Verb** | 77.01 | 0.993 | 0.798 |
| **Object** | 66.53 | 0.995 | 0.84 |
| | **Layer 5** | | |
| **CLS** | 18.64 | 0.999 | 0.996 |
| **Subject** | 73.40 | 0.994 | 0.84 |
| **Verb** | 82.05 | 0.992 | 0.804 |
| **Object** | 71.46 | 0.995 | 0.858 |
| | **Layer 6** | | |
| **CLS** | 21.88 | 0.999 | 0.985 |
| **Subject** | 78.51 | 0.993 | 0.85 |
| **Verb** | 85.13 | 0.992 | 0.72 |
| **Object** | 73.38 | 0.994 | 0.826 |

Table 8: Compositionality for Different Words in a Sentence (layer 7 to layer 12)

| Metric | Values | | |
|---|---|---|---|
| | **L2 Loss** | **Cosine Similarity** | **Retrieval Accuracy** |
| **Layer 7** | | | |
| **CLS** | 34.64 | 0.998 | 0.981 |
| **Subject** | 72.49 | 0.994 | 0.826 |
| **Verb** | 84.36 | 0.991 | 0.717 |
| **Object** | 70.17 | 0.995 | 0.835 |
| **Layer 8** | | | |
| **CLS** | 25.48 | 0.999 | 0.959 |
| **Subject** | 62.49 | 0.994 | 0.821 |
| **Verb** | 74.51 | 0.992 | 0.685 |
| **Object** | 62.28 | 0.995 | 0.85 |
| **Layer 9** | | | |
| **CLS** | 30.22 | 0.997 | 0.949 |
| **Subject** | 56.14 | 0.994 | 0.865 |
| **Verb** | 68.16 | 0.991 | 0.793 |
| **Object** | 54.89 | 0.994 | 0.887 |
| **Layer 10** | | | |
| **CLS** | 54.01 | 0.993 | 0.98 |
| **Subject** | 66.86 | 0.992 | 0.951 |
| **Verb** | 86.81 | 0.987 | 0.922 |
| **Object** | 64.43 | 0.993 | 0.974 |
| **Layer 11** | | | |
| **CLS** | 76.66 | 0.988 | 0.996 |
| **Subject** | 72.46 | 0.991 | 0.987 |
| **Verb** | 96.29 | 0.985 | 0.976 |
| **Object** | 79.55 | 0.990 | 0.992 |
| **Layer 12** | | | |
| **CLS** | 103.91 | 0.983 | 0.995 |
| **Subject** | 96.46 | 0.985 | 0.988 |
| **Verb** | 108.58 | 0.980 | 0.993 |
| **Object** | 102.55 | 0.983 | 0.991 |

