# OpenReview forum: "Understanding Compositionality in Data Embeddings"
_TMLR — Rejected by TMLR_

### Review · Reviewer_ELjJ · 2024-03-22

**Summary Of Contributions:**

This paper investigates embeddings with regards to the encoded information, and the degree to which embeddings can be treated as the definition of multiple types of information.
In total, two methods for the compositionality of embeddings: Correlation-based Compositionality Detection, Additive Compositionality Detection.
These methods are applied to three embedding types: word embeddings, sentence embeddings and knowledge graph embeddings.

**Audience:**

Yes

**Broader Impact Concerns:**

Not relevant for this paper

**Claims And Evidence:**

No

**Requested Changes:**

Request changes
- Provide a motivation for the choice of models, evaluation tasks, and two methods
- Potentially consider the use “modern” embeddings/language models
-Better distinguish between debiasing and compositionality. At times it sounds like the paper is introducing a debiasing method, at other times it mentions it as future work (there is the option to add a future work section and add debiasing details there).
- Usage of more advanced/complex benchmarking methods (either existing publications or other ones than random shuffle). Potentially update findings/significance of results. Currently, results show that it is better than random, which does not mean it generalises or works or is particularly good
- Provide more details on SBERT (how sentence embeddings are created) and outline why the experimental task is valid and differs from the procedure to create them. Potentially use a different, existing task for evaluation.
- Page 3: can existing approaches be applied to other types of embeddings?
- This could be a phrasing/clarification issue: It sounds like that for knowledge graphs, the embeddings are calculated from the user information matrix rather than that embeddings are decomposed into the user matrix.


Minor changes:
- Abstract makes general claims about composability. It could be written less general, taking the investigated models and tasks in account
- Introduction: Potentially include a Figure to explain knowledge graphs and embeddings
- Update figure 2. It shows graphs for each of the embeddings
- Page 7: Figure ??
- Beginning of Section 2.2 is vague
- 2.2.3 speaks a lot about bias and not knowledge graphs
- Page 12: “Alogorithm”
- 4.1: Unclear if static embeddings can encode syntactic information
- 4.1.4 unclear what shuffling is doing
- 4.3 Konwledge
- 4.3.3 what are the 14 groups and why are they chosen
- 4.3.4 what are the 241 groups

**Strengths And Weaknesses:**

Strengths
- The investigation considers three different embedding types
- The context and background is very exhaustive
- Detailed investigation of each of the three experiments, with details on datasets, evaluation and statistical tests


Weaknesses
- The embeddings used and considered related work are comparably old. Word2Vec is from 2013 and BERT is not advanced in comparison to the vast amount of large language models published. Moreover, these benefit from taking advantage of context and not representing each word by a single vector.
- The SBERT evaluation task appears related to how the sentence embeddings are created (mean over each word)
- It is unclear whether the performance is good when only compared against a random baseline

---

### Review · Reviewer_VAT4 · 2024-03-25

**Summary Of Contributions:**

This paper analyzes Word2Vec embeddings, BERT sentence embeddings, and knowledge graph embeddings, revealing that they can be approximated as linear compositions of interpretable features. Word2Vec embeddings can be approximated with either semantic or morphological features, sentence embeddings can be approximated as a weighted sum of words, and knowledge graph embeddings can be (problematically) approximated as a combination of demographic attributes.

**Audience:**

Yes

**Claims And Evidence:**

Yes

**Requested Changes:**

See Strengths and Weaknesses for elaborations on these points.

Critical: Toning down claims substantially. Engagement with prior work. Adding some more embedding types from more modern architectures/models.

Recommended: Writing revisions for concision and clarity. Demonstrating more complete additive reconstruction or more surprising features encoded in any of the embeddings.

**Strengths And Weaknesses:**

Strengths:
This work presents two methods for assessing compositionality within embeddings. The authors demonstrate that their main results hold across both methods. Additionally, the authors explore 3 types of embeddings, testing straightforward hypotheses for each embedding.

Weaknesses:
The main claims in this paper are quite overstated. For example, in the BERT sentence embedding section, the authors state “These results show that the BERT sentence embedding can be decomposed into three separate components: subject, verb, and object. And those components can then be used to predict the embedding of a new sentence.“ Except this is only true in comparison to the random permutation control that the authors consider. The average loss for these reconstructions is phenomenally high, at nearly 100! Clearly, the embeddings are not merely compositions of these three independent words, but instead are quite contextualized (which is the point of using BERT sentence embeddings in the first place). This critique holds generally for all results: achieving lower loss than a random feature permutation merely rejects the null hypothesis that these features are not meaningfully represented within an embedding. It does not mean that the embeddings can be meaningfully decomposed into (just) these features.

Because of this, the experimental results become almost trivial. It is clear that a sentence embedding is going to contain some information about the constituent words, and that word embeddings are going to contain some semantic information. Also, a broad range of research has demonstrated that demographic properties are decodable from embeddings (though perhaps this property is under-studied in knowledge graph embeddings). The results would be more interesting and useful for the community if either (1) the embedding reconstruction loss was lower, (2) the properties encoded in the embedding were more surprising, (3) more types of embeddings were investigated, and systematic differences between them were described. Some relevant comparisons might be embeddings from the [CLS] token of a standard BERT vs. embeddings from SBERT, or embeddings from RoBERTa vs. BERT (as RoBERTa lacks a next sentence prediction objective), or embeddings from BERT vs. GPT, or embeddings from BERT early in training vs. later in training (which can be accomplished with the MultiBERTs model releases), or small vs. large pythia models. Finding differences between these types of embeddings would be legitimately useful to the broader community. Instead, many of the analyses that this paper presents are done using simple attributes and outdated methods for generating embeddings.

Additionally, it is crucial that the authors more thoroughly engage with prior work on compositionality in embeddings. As a start, here is a list of relevant papers:
Soulos, Paul, et al. "Discovering the compositional structure of vector representations with role learning networks." arXiv preprint arXiv:1910.09113 (2019).
Yu, Lang, and Allyson Ettinger. "Assessing phrasal representation and composition in transformers." arXiv preprint arXiv:2010.03763 (2020).
Ettinger, Allyson, et al. "Assessing composition in sentence vector representations." arXiv preprint arXiv:1809.03992 (2018).
Klafka, Josef, and Allyson Ettinger. "Spying on your neighbors: Fine-grained probing of contextual embeddings for information about surrounding words." arXiv preprint arXiv:2005.01810 (2020)
Lepori, Michael A., and R. Thomas McCoy. "Picking BERT's brain: Probing for linguistic dependencies in contextualized embeddings using representational similarity analysis." arXiv preprint arXiv:2011.12073 (2020).
Chrupała, Grzegorz, and Afra Alishahi. "Correlating neural and symbolic representations of language." arXiv preprint arXiv:1905.06401 (2019).
John Wu, Yonatan Belinkov, Hassan Sajjad, Nadir Durrani, Fahim Dalvi, and James Glass. 2020. Similarity analysis of contextual word representation models. In Proceedings of the 58th Annual Meeting of the Association for Computational Linguistics, pages 4638–4655, Online, July. Association for Computational Linguistics.
Kim, Najoung, and Tal Linzen. "Compositionality as directional consistency in sequential neural networks." Workshop on Context and Compositionality in Biological and Artificial Neural Systems. 2019
Tenney, Ian, et al. "What do you learn from context? probing for sentence structure in contextualized word representations." arXiv preprint arXiv:1905.06316 (2019).

More general work on compositionality in neural networks should be engaged with in some capacity:
Lake, Brenden, and Marco Baroni. "Generalization without systematicity: On the compositional skills of sequence-to-sequence recurrent networks." International conference on machine learning. PMLR, 2018
Kim, Najoung, and Tal Linzen. "COGS: A compositional generalization challenge based on semantic interpretation." Proceedings of the 2020 Conference on Empirical Methods in Natural Language Processing (EMNLP). 2020.
Lake, Brenden M., and Marco Baroni. "Human-like systematic generalization through a meta-learning neural network." Nature 623.7985 (2023): 115-121.
Hupkes, Dieuwke, et al. "Compositionality decomposed: How do neural networks generalise?." Journal of Artificial Intelligence Research 67 (2020): 757-795.
Lepori, Michael, Thomas Serre, and Ellie Pavlick. "Break it down: Evidence for structural compositionality in neural networks." Advances in Neural Information Processing Systems 36 (2024).

Additionally, the writing in this paper can be made substantially more concise. There are several sections that contain redundant information. On the other hand, some concepts are under-explained/hard to follow, notably the correlation based compositionality detection explanation. Finally, the example of “compositionality” being a composition of “com”+”pos” + “ition” + “ality” is somewhat strange, and could easily be replaced by a more natural example.

---

### Review · Reviewer_WP56 · 2024-04-16

**Summary Of Contributions:**

This paper studied the compositionality of embeddings in word embeddings, sentence embeddings, and knowledge graph embeddings. Although compositionality had previously been studied, the paper argued that no method existed to provide quantitative, statistical tests. The paper proposed two compositionality detection methods based on the subject embeddings' projections and attribute one-hot vectors. The projects were trained using L2-norm (additive) and cosine similarity (correlational).

In the experiments, the paper presented that for all applications of embeddings, including word embeddings, sentence embeddings, and knowledge graph embeddings, there was a learned project that decomposed embedding into a linear system and was significantly better than the permuted ones.

**Audience:**

Yes

**Broader Impact Concerns:**

There was no ethical concern.

**Claims And Evidence:**

No

**Requested Changes:**

### Important

1. I found the kernel perspective of two embeddings interesting, but it was not quite relevant to the proposed method. Instead of discussing the kernel perspective, I think the authors should pay more attention to how the representation learning method is related to compositionality and provide more context for previous work on compositionality.
2. The authors should revise sections relevant to the sentence embedding and private attributes to clear the above confusion.
3. Upon reviewing relevant literature, I found an IDAS paper with a similar title and abstract. Please justify this manuscript.

### Improvement

1. Adding experiment results for word senses would widen its impact to confirm/reject previous work on multi-sense embedding.
2. Adding more sentence variety to the sentence embedding experiments or justifying why subject-verb-object was sufficient to conclude the compositionality.
3. Revising Section 2.1.3 for its relevancy to the rest of the paper. There was no point in discussing Eq 3 to 6 if the rest of the paper did not contrast or analyze them.

**Strengths And Weaknesses:**

### Strengths

1. The paper studied a property of embedding -- fundamental objects that most models relied upon.
1. The test showed quantitatively that the compositionality existed in word, sentence, and knowledge graph embeddings.

### Weaknesses

1. The sentence embedding result seemed rather insignificant, and the discussion of it was confusing.
    1. The transformer generates a sentence embedding by the words' attention mechanism (albeit multi-headed), so it was natural that the sentence embedding would be the sum of its embedding. However, the output embeddings of any word are produced through the same mechanism. This raised a more interesting question of whether they also capture the same compositionality and a better baseline than random permutation.
    1. The mode of sentence compositionality studied in this paper was too simple. If the goal was to show an additive compositionality of its words, the experiment was not restricted to subject, verb, and object.
    1. It wasn't very clear in Section 4.2.3 and Section 5. The sections discussed the cosine similarity of the sentence embeddings in percentage (not sure why?), the discussion (Section 5) using the permutation performance (why not $\hat{U}$?), and how to derive 64% similarity.
1. The compositionality of word embedding has been studied extensively. The author might want to review prior models that proposed the compositionality of senses, in addition to hypernyms and morphology ([Gittens et al., 2017](https://aclanthology.org/P17-1007.pdf), [Seonwoo et al., 2019](https://aclanthology.org/D19-5551.pdf)).
1. The paper claimed that "... can be decomposed into a sum of embedding corresponding to demographic attributes ..., not used in the training ..." However, the experiments used the attributes to learn the projection. It was, then, unclear how the evidence supported the claim.

---

### Decision · Action_Editor_23Qd · 2024-05-28

**Recommendation:** Reject

**Comment:**

Reviewers appreciated the topic of the manuscript (WP56) and the quantitative assessment of the additive compositionality across the two proposed methods (WP56, VAT4), the exploration of three embedding types for the study (VAT4, ELjJ), and the provided details and context (ELjJ).

At the same time, they raised several concerns, mostly regarding (i) the choice of random permutations as a unique alternative for hypothesis testing (i.e., VAT4, ELjJ, WP56) making it unclear whether another type of compositions (even from previous work) may better explain the data; (ii) the embeddings considered, not including more recent ones (i.e., ELjJ, VAT4), discussion of previous work (WP56, VAT4), and the lack of insights on the obtained results (WP56, e.g., characteristics of the linear system, why this compositionality emerges).  Moreover, they raised concerns on presentation issues (VAT4 e.g., typos, redundancies) and on the overlap with similar work (WP56).

The responses managed to address some of the concerns (e.g., why random permutation supports the result, and why the obtained values are significant), better presenting the findings and previous work. At the same time, two main issues remain:
1. Experiments. the use of random permutation as the main counterpart was still considered limited in the final assessment of the strength of the obtained results, as well as the lack of more embedding types.

2. Overlap with previous work. The responses recognized that the article is a (substantially) extended version of a previous IDA work [1] (WP56). Moreover, there is also a focus on debiasing (ELjJ) which is the main topic of another work [2] which shares some part with the current submission. As an example, the methods presented in [2] for debasing and their examples (Fig. 2 and 3 of [2]) are reported in this submission (Fig. 3 and 4). While this is not a problem per se for journals (whenever the differences are highlighted) it is for TMLR as, per regulation, it does not accept extensions and re-use of materials but only original work (https://jmlr.org/tmlr/editorial-policies.html). The current version of the manuscript violates this rule.

3. Finally, the writing should be also improved as typos and formatting issues have not been fully addressed (e.g., Table ?? on Page 17).

While the last point can be easily addressed, the first would require more analyses and the second is a major one for this journal. Therefore, the AE recommends the rejection of the current version of the work. Nevertheless, as the additive compositionality test is interesting, a major revision of the work (addressing the overlap issues with previous work and including more analyses) should be considered for this journal.

**References**:\
[1] Xu, Z., Guo, Z., & Cristianini, N. On compositionality in data embedding. In International Symposium on Intelligent Data Analysis (pp. 484-496). Cham: Springer Nature Switzerland.\
[2] Guo, Z., et al. "EXTRACT: Explainable Transparent Control of Bias in Embeddings." First AEQUITAS Workshop on Fairness and Bias in AI, co-located with ECAI 2023.

**Audience:**

Studying data embedding properties is a direction of interest to all researchers working on data representations. Providing statistical evidence for additive compositionality poses interesting basis for further investigations.

**Claims And Evidence:**

This work studies the compositionality of data embeddings (e.g., word, sentence, knowledge graph) and their composition as a linear composition of their parts (e.g., morphological information, words, attributes). The paper proposes two methods for detecting this i.e., correlation-based (studying the correlation between data and attributes) and additive (treating embeddings as the sum of meaningful attributes).

The hypothesis of additive compositionality is tested against random words/part permutations, providing clear statistical evidence that additivity explains data better than random chance. Reviewers reported mixed opinions about this choice, some considering it to well support the evidence and others as a too simple baseline. There have been also criticisms on the type of embeddings considered and the lack of more recent ones to better support the claims and make them more general.

**Resubmission Of Major Revision:**

The authors may consider submitting a major revision at a later time.